# Predictive Values of Hematological Parameters for Determining Imminent Brain Death: A Retrospective Study

**DOI:** 10.3390/medicina59020417

**Published:** 2023-02-20

**Authors:** Eralp Çevikkalp, Mevlüt Özgür Taşkapılıoğlu

**Affiliations:** 1Departments of Anaesthesiology and İntensive Care, Bursa City State Hospital, 16110 Bursa, Turkey; 2Departments of Neurosurgery, Bursa Uludağ University, 16059 Bursa, Turkey

**Keywords:** lymphocyte-to-monocyte ratio, neutrophil-to-lymphocyte ratio, platelet-to-lymphocyte ratio, platelet-to-neutrophil ratio, brain death

## Abstract

*Background and Objectives*: The inflammatory cells released after intracranial hemorrhage, such as monocytes, macrophages, and neutrophils, activate the inflammatory system. These parameters can be used to evaluate the clinical course of diseases. This study aims to evaluate these parameters as possible predictors for evaluating the development of brain death. *Materials and Methods:* Patients with a Glasgow coma scale score below 7 were assigned to Group BD (patients with brain death) and Group ICH (intracranial hemorrhage). The neutrophil, lymphocyte, platelet, monocyte counts, neutrophil-to-lymphocyte ratio (NLR), platelet-to-lymphocyte ratio (PLR), lymphocyte-to-monocyte ratio (LMR), and platelet-to-neutrophil ratio (PNR) were measured at admission. *Results*: A high WBC count, neutrophil count, NLR, and PLR and a low lymphocyte count, LMR, and PNR were found to be significant for determining brain death. The area under the curve (AUC) values of NLR, PNR, PLR, and LMR to discriminate brain death were 0.63, 0.61, 0.56, and 0.61, respectively. *Conclusion:* NLR, PNR, PLR, and LMR are easily accessible and reliable parameters that can be used to predict the development of brain death and can be estimated by a simple complete blood count test.

## 1. Introduction

The diagnosis of brain death is carried out by clinical and radiological confirmation of irreversible damage to all brain functions. Neurological tests demonstrating the absence of brain stem functions, intracranial pathology that may cause coma, and apnea test positivity are required to establish the diagnosis of brain death [1]. Mechanically ventilated and deeply comatose patients admitted to the intensive care unit with a Glasgow coma scale score of 3 and absence of at least 3 or more brainstem reflexes or a Full Outline of UnResponsiveness (FOUR) score of 0 due to irretrievable brain damage are considered to fulfill the definition of “imminent brain death” for potential donor candidates [2].

A stroke is defined as a loss of brain functions due to an abnormal blood supply. An ischemic stroke is a cerebrovascular accident resulting from impaired blood supply to the brain, while intracranial hemorrhage is defined as bleeding into the brain parenchyma [3,4]. Post-stroke brain damage is primarily caused by the mass effect related to the intracerebral event and the secondary effect of the neurotoxicity of blood components, followed by intracerebral cellular and inflammatory activation, disruption of the blood–brain barrier, activation of microglia/macrophages in the surrounding brain parenchyma, neutrophil and monocytes influx, cytokine release, and activation of the complement cascade [5,6].

The evaluation of post-stroke inflammatory parameters can be used to predict the clinical course of patients. A complete blood count is a simple method frequently used in the monitoring of inflammatory parameters. The neutrophil-to-lymphocyte ratio (NLR), lymphocyte-to-monocyte ratio (LMR), and platelet-to-lymphocyte ratio (PLR) assessed using a complete blood count have been shown to be a predictive value for determining the prognosis of stroke patients [7,8].

Brain-dead patients are potential donor candidates for organ transplantation. Therefore, these patients should be followed up meticulously for imminent brain death and should be evaluated early in order to increase the quality of organ perfusion during the process. The aim of this study is to investigate whether NLR, LMR, PLR, and platelet-to-neutrophil ratio (PNR) have an early predictive value for evaluating imminent brain death.

## 2. Materials and Methods

After obtaining the approval of Uludağ University Faculty of Medicine Non-Interventional Research Ethics Committee (decision no: 2011-KAEK-26/165), the study included patients admitted to the anesthesia intensive care unit of Private Medicabil hospital for intracranial bleeding with a Glasgow coma score below 7 between 01 January 2019 and 01 February 2022. The patients were divided into two separate groups: Group BD (patients with brain death) and Group ICH (patients with intracranial hemorrhage without brain death). In Group BD, according to the Ministry of Health regulation for organ and tissue transplantation, the absence of brainstem reflexes, a coma status that has been confirmed not to be induced by hypothermia or drugs, and a positive apnea test result were evaluated in favor of the diagnosis of brain death. To perform the apnea test, the patient was disconnected from the mechanical ventilator and administered intratracheal oxygen to prevent the auto-triggering of cardiac origin after providing normothermia, normotension, normovolemia, and PaCO_2_ of 35 to 45 mm Hg and PaO2 of >200 mm Hg. The apnea test was considered positive if spontaneous breathing was absent despite a PaCO_2_ value ≥60 mm Hg and/or an increase in PaCO_2_ of 20 mm Hg or more from baseline at the end of the test. The patients underwent cerebral CT angiography to confirm brain death and to detect the absence of intracranial blood flow. The patients without intracranial blood flow were considered brain dead [1]. The patients who died in the intensive care unit due to intracranial hemorrhage and who did not have brain death were included in the ICH group. The patients who had sequelae of a previous cerebrovascular accident, 24 h after the onset of the cerebrovascular accident, patients with a history of infection 2 weeks before hospitalization or with signs of infection during hospitalization, patients with malignancy, immunodeficiency, immunotherapy, or severe coronary artery disease, and patients who died due to secondary infection were not included in the study. The blood parameters of neutrophil, lymphocyte, platelet, and monocyte analyzed to evaluate the condition of patients at admission to the intensive care unit were recorded. The patients’ demographic data and indications for hospitalization were recorded as well.

### Statistical Analysis

The Shapiro–Wilk test was used to check whether continuous variables followed a normal distribution. The continuous variables were expressed with median (minimum: maximum) values, while the categorical variables were expressed as n (%). For continuous variables, the Mann–Whitney U test was used to compare groups. The categorical variables were analyzed using the chi-square test, Fisher’s exact chi-square, and Fisher-Freeman-Halton tests. The receiver operator characteristic (ROC) curve analyses were carried out to estimate the sensitivity and specificity of white blood cell (WBC), neutrophil, lymphocyte counts, NLR, PNR, PLR, and LMR for predicting brain death. SPSS software (IBM Corp. Released 2012. IBM SPSS Statistics for Windows, Version 26.0. Armonk, NY, USA: IBM Corp.) was used for statistical analysis, and the type I error rate was set at 5%.

## 3. Results

A total of 302 patients were included in the study. Ten patients were excluded due to incomplete and missing data, and 11 patients were excluded because they did not meet the inclusion criteria. The analysis included 281 patients. Of the patients included in the analysis, 136 were female and 145 were male. Group BD included 128 patients, while Group ICH included 153 patients. There was no difference between the groups in terms of age, gender distribution, and chronic diseases (*p* = 0.820) (Table 1).

The median WBC and neutrophil count were higher in Group BD (*p* = 0.002, *p* = 0.004). The median lymphocyte count was higher in Group ICH (*p* = 0.021). There was no difference between the groups in terms of PLT and monocyte count (*p* = 0.393, *p* = 0.239). The median NLR and median PLR were higher in Group BD (*p* < 0.001, *p* = 0.046). The median LMR and median PNR were higher in Group ICH (*p* = 0.006, *p* = 0.002) (Table 2).

An ROC analysis was performed to determine the sensitivity and specificity of the WBC count for predicting brain death (Figure 1). The area under the ROC curve (AUC) was estimated as 0.60 for a WBC count >13,760 cells/mm^3^ (sensitivity 53.91%, specificity 70.59%, and *p* = 0.002) and a WBC count >13,760 cells/mm^3^ was significantly associated with an increased risk of brain death (Table 3).

The ROC analysis performed to estimate the sensitivity and specificity of neutrophil count for predicting brain death (Figure 1) showed an AUC value of 0.60 for a neutrophil count >12,270 cells/µL (sensitivity 50.78%, specificity 68.63%, and *p* = 0.003); a neutrophil count >12,270 cells/µL was significantly associated with an increased risk of brain death.

According to the ROC analysis performed to determine the sensitivity and specificity of the lymphocyte count for predicting brain death (Figure 1), the AUC value was 0.58 for a lymphocyte count ≤810 cells/µL (sensitivity 42.97%, specificity 73.20%, and *p* = 0.021); a lymphocyte count ≤810 cells/µL was significantly associated with an increased risk of brain death (Table 3).

The ROC analysis carried out to calculate the sensitivity and specificity of NLR for predicting brain death (Figure 1) revealed an AUC value of 0.63 for NLR >12.9, (sensitivity 55.47%, specificity 71.90%, and *p* < 0.001); a NLR >12.9 was significantly associated with an increased risk of brain death (Table 3).

In the ROC analysis performed to determine the sensitivity and specificity of PNR for predicting brain death (Figure 1), the AUC value was estimated as 0.61 for PNR ≤22.27 (sensitivity 69.53%, specificity 50.33%, and *p* = 0.001); a PNR ≤22.27 was significantly associated with an increased risk of brain death (Table 3).

The ROC analysis carried out to estimate the sensitivity and specificity of PLR for predicting brain death (Figure 1) showed an AUC value of 0.56 for PLR >287.36 (sensitivity 39.06%, specificity 74.51%, and *p* = 0.045); a PLR >287.36 was significantly associated with an increased risk of brain death (Table 3).

According to the ROC analysis performed to determine the sensitivity and specificity of LMR for predicting brain death (Figure 1), the estimated AUC value was 0.61 for LMR ≤2.63 (sensitivity 59.84%, specificity 59.48%, and *p* = 0.005); a LMR ≤2.63 was significantly associated with an increased risk of brain death (Table 3).

## 4. Discussion

This study investigated whether hematological inflammatory parameters may be predictors of brain death and demonstrated that high WBC, neutrophil, NLR, and PLR and low lymphocyte, PNR, and LMR values can be relevant predictors of brain death.

Inflammatory processes begin in the body after mediators are released from damaged tissues following intracranial hemorrhage. Depending on the severity of the processes, serious clinical results and organ damage are observed in patients. The early detection of progression to brain death for cadaveric organ donation is important for improving organ perfusion.

Neuroendocrine substances released after intracranial hemorrhage induce an increase in the leukocyte count. There are studies showing the association of leukocytosis with severe disability, poor neurological outcomes, and mortality [9,10,11]. The INTERACT2 study showed an association between high admission WBC values and mortality [12]. A meta-analysis found that high admission WBC values were associated with short-term mortality and poor neurological outcomes [9]. Another study, which associated high leukocyte values with short-term mortality, found an AUC value of 0.642 for WBC >9400 cells/mm^3^ [10]. The results of our study demonstrated higher WBC values in Group BD, with an estimated AUC value of 0.60 for WBC >13,760 cells/mm^3^ (sensitivity 53.91%, specificity 70.59%, *p* = 0.002), which are similar to those reported in the literature.

It has been shown that high NLR values after intracranial hemorrhage are associated with poor prognosis and in-hospital mortality. A study of 1000 patients reported high NLR in patients who died in the hospital with a diagnosis of intracranial hemorrhage [12]. A study examining complete recovery after subarachnoid hemorrhage found high NLR in patients with poor neurological outcomes [13]. The NLR values were found to be low in patients with good short-term neurological outcomes after intracranial hemorrhage [14]. Tao et al. showed that NLR >6.62 was associated with high 90-day mortality after intracranial hemorrhage. This study found a median NLR value of 13,600 in patients who died in the hospital, which was twice the value of survivors [15]. In our study, the estimated median value of Group BD is similar to that in the literature. Our results showed an increased risk of developing brain death for NLR >12.9.

The neutrophil, lymphocyte, and platelet values measured at admission after subarachnoid hemorrhage increase the risk of in-hospital mortality [16]. In cases where the neutrophil count is above 11,590 cells/mm^3^, the AUC value was measured as 0.67 [16]. In the patients with intracranial hemorrhage, a high admission neutrophil count and low admission lymphocyte count increased mortality and are associated with poor neurological outcomes [15,17]. In our study, Group BD had a low lymphocyte count and a high neutrophil count. A neutrophil count >12,270 cells/mm^3^ and a lymphocyte count <810 cells/mm^3^ increase the risk of brain death. Our study revealed similar results to those reported in the literature.

High PLR and low PNR after aneurysmal hemorrhage have been shown to be associated with poor neurological outcomes [18]. High PLR values during admission to the intensive care unit are associated with poor early neurological outcomes [19]. A study investigating long-term neurological outcomes in patients followed up for an acute ischemic stroke showed that low PLR and LMR were associated with poor prognosis [20]. Our study demonstrated that high PLR and low PNR and LMR were effective in predicting brain death.

Our study has some limitations. First, patients are admitted to our intensive care unit from other centers through the emergency referral unit, so the admitted patients may not reflect the general population. Second, the data of the patients were not followed up dynamically because the study had a retrospective design. It may be possible to obtain different results in further studies for the determined values.

In conclusion, NLR, PNR, PLR, and LMR are easily accessible and reliable parameters that can be used to predict the development of brain death and can be calculated by performing a simple complete blood count test.

## Figures and Tables

**Figure 1 medicina-59-00417-f001:**
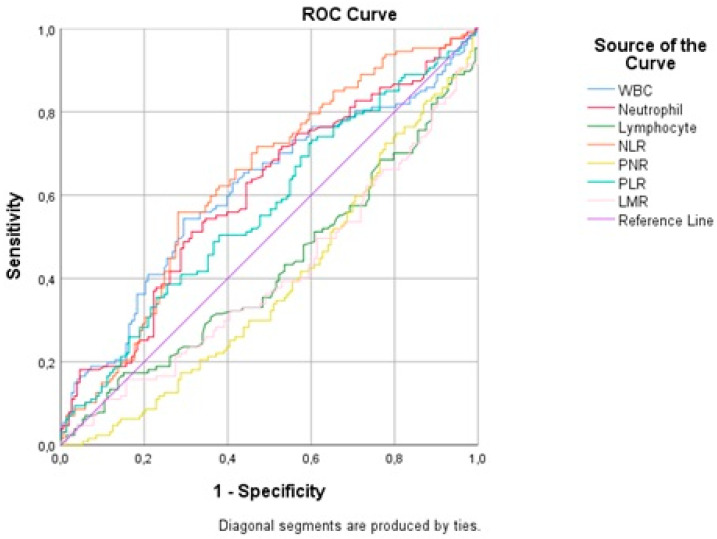
ROC curve analysis.

**Table 1 medicina-59-00417-t001:** Demographic characteristics and comorbidities.

	n	Brain Death	n	Hemorrhage	*p*-Value
**Age (years)**	128	63.50 (19:94)	153	66 (18:91)	0.271 ^a^
**Gender**					
*Female*	128	61 (47.70%)	153	75 (49%)	0.820 ^b^
*Male*		67 (52.30%)		78 (51%)	
**Comorbidities**					
*HT*	128	87 (68%)	153	96 (62.70%)	0.054 ^b^
*DM*	128	28 (21.90%)	153	28 (18.30%)	0.455 ^b^
*COPD*	128	9 (7%)	153	4 (2.60%)	0.079 ^b^
*CRF*	128	2 (1.60%)	153	0	0.207 ^c^
*CAD*	128	15 (11.70%)	153	14 (9.20%)	0.481 ^b^
*AF*	128	3 (2.30%)	153	0	0.093 ^c^

HT: Hypertension, DM: Diabetes Mellitus, COPD: Chronic Obstructive Pulmonary Disease, CRF: Chronic Renal Failure, CAD: Coronary Artery Disease, AF: Atrial Fibrillation; Variables are expressed with median (minimum:maximum) and n (%) values; ^a^: Mann–Whitney U Test, ^b^: Pearson’s Chi-square Test, ^c^: Fisher’s Exact Test.

**Table 2 medicina-59-00417-t002:** Comparison of hematological measurements.

	n	Brain Death	n	Hemorrhage	*p*-Value
WBC	128	14,240 (3400:79,800)	153	11,650 (2580:31,310)	**0.002 ^a^**
PLT	128	216,500 (16,000:480,000)	153	223,000 (9000:500,000)	0.393 ^a^
Neutrophil	128	12,360 (1890:66,540)	153	9940 (1580:29,550)	**0.004 ^a^**
Lymphocyte	128	980 (0.83:23,550)	153	1180 (220:12,540)	**0.021 ^a^**
Monocyte	128	395 (0:6180)	153	350 (10:1160)	0.239 ^a^
Neutrophil/Lymphocyte	128	13.77 (0.34:19,542.17)	153	8.66 (0.42:98)	**<0.001 ^a^**
PLT/Neutrophil	128	17.59 (0.24:57.91)	153	22.29 (1.81:143)	**0.002 ^a^**
PLT/Lymphocyte	128	242.13 (3.33:238,554.22)	153	218 (3.52:925.93)	**0.046 ^a^**
Lymphocyte/Monocyte	127	2.27 (0.25:142)	153	3.11 (0.64:68)	**0.006 ^a^**

WBC: White Blood Cell, PLT: platelet; Variables are expressed with median (minimum:maximum) values; ^a^: Mann–Whitney U Test.

**Table 3 medicina-59-00417-t003:** Sensitivity and specificity of WBC, neutrophil, lymphocyte, NLR, PLR, and LMR for predicting brain death.

	AUC	95% (CI)	Optimal Cut-Off Point	Sensitivity	Specificity
WBC	0.60	(0.551:0.668)	>13,760	53.91%	70.59%
Neutrophil	0.60	(0.545:0.662)	>12,270	50.78%	68.63%
Lymphocyte	0.58	(0.517:0.636)	<810	42.97%	73.20%
NLR	0.63	(0.576:0.692)	>12.9	55.47%	71.90%
PLR	0.56	(0.507:0.626)	>287.36	39.06%	74.51%
LMR	0.59	(0.536:0.654)	≤2.63	59.84%	59.48%
PNR	0.61	(0.55:0.66)	≤22.27	69.53%	50.33%

WBC: White Blood Cell, NLR: Neutrophil-to-Lymphocyte Ratio, PLR: Platelet-to-Lymphocyte Ratio, LMR: Lymphocyte-to-Monocyte Ratio, PNR: Platelet-to-Neutrophil Ratio.

## Data Availability

The data presented in this study are available on reasonable request from the corresponding author.

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
