# Peer review of "Predictive Values of Hematological Parameters for Determining Imminent Brain Death: A Retrospective Study"

_medicina, 2023, doi:10.3390/medicina59020417_

Round 1
Reviewer 1 Report
1. The title should mention the article type. E.g. cross-sectional, retrospective…
2. Apparently, there are other studies in the literature with the same aim as the authors. Could the authors explain how their manuscript differs from others in the literature?
3. There are some grammatical errors throughout the manuscript. The article should follow American or British English.
4. How do the authors compare individuals with BD and intracranial hemorrhages? Could the authors explain the basis for this comparison? Are there other studies with this similar comparative analysis?
5. How were confounding variables assessed?
I selected a major revision based on the fact that the authors should answer the following: "2. Apparently, there are other studies in the literature with the same aim as the authors. Could the authors explain how their manuscript differs from others in the literature?" This query does not have an answer based on my previous knowledge. The authors performed a work that does not present anything new. The only thing that they can say that is unique is the fact that the study was done in a specific population. E.g., Turkey. Another point for significant revision is the "4. How do the authors compare individuals with BD and intracranial hemorrhages?" I could not understand how the authors could compare individuals with different pathologies for the same outcome. The groups were not correctly divided. These different conditions have well-known different outcomes. It is almost impossible to exclude confounding factors of both pathologies.Author Response
- The title should mention the article type. E.g. cross-sectional, retrospective…
Thanks for your comment. Article title has been changed after your recommendation.
- Apparently, there are other studies in the literature with the same aim as the authors. Could the authors explain how their manuscript differs from others in the literature?
We would like to thank for this comment. In our study, patients with GCS of 7 and below, who were followed up for possible brain death, were included. The patients who died in the intensive care unit were divided into Group BD and ICH. In our study, it was aimed that hematological parameters may be effective in predicting brain death. We think that we make an additional contribution to the literature by predicting possible brain death and comparing hematological parameters evaluated in the early period during hospitalization.
The fallowing paragraphs ‘’ In Group BD According to the Ministry of Health regulation for organ and tissue transplantation, the absence of brainstem reflexes, coma status that has been con-firmed not to be induced by hypothermia or drugs, and a positive apnea test result were evaluated in favor of the diagnosis of brain death. To perform the apnea test, the patient was disconnected from the mechanical ventilator and given intratracheal oxy-gen to prevent auto-triggering of cardiac origin after providing normothermia, normo-tension, normovolemia, and PaCO2 of 35 to 45 mm Hg and PaO2 of >200 mm Hg. The apnea test was considered positive if spontaneous breathing was absent despite a Pa-CO2 value ≥60 mm Hg and/or an increase in PaCO2 of 20 mm Hg or more from base-line at the end of the test. Patients underwent cerebral CT angiography to confirm brain death and to detect the absence of intracranial blood flow. Patients without intracranial blood flow were considered brain dead [1]. Patients who died in the inten-sive care unit due to intracranial hemorrhage and who did not have brain death were included in the ICH group.’’ Added to the material and method section. The sentence ‘’ patients who died due to secondary infection ‘’ added in to exclucion criteria.
- There are some grammatical errors throughout the manuscript. The article should follow American or British English.
The manuscript was checked by a native speaker because of limited time to resubmission. The English will be edited by a professional editor after the desicion.
- How do the authors compare individuals with BD and intracranial hemorrhages? Could the authors explain the basis for this comparison? Are there other studies with this similar comparative analysis?
We would like to thank for this valuable comment. The situation caused by the lack of explanation has been corrected. In our study, the groups were arranged by dividing the patients who were followed in terms of possible brain death and those who died into two groups. We think that there are similar groups in the evaluation of hematological parameters in predicting brain death in patients who were admitted to the intensive care unit and died with intracranial bleeding.
- How were confounding variables assessed?
When a confounting situation was encountered during the measurements, the opinion of an experienced radiologist was consulted.

Reviewer 2 Report
No

Author Response
Thanks for the reviewer. we fixed mistakes that the referee suggested to be corrected.